# Impact of 1,8-Diiodooctane (DIO) Additive on the Active Layer Properties of Cu_2_ZnSnS_4_ Kesterite Thin Films Prepared by Electrochemical Deposition for Photovoltaic Applications

**DOI:** 10.3390/ma16041659

**Published:** 2023-02-16

**Authors:** Elmoiz Merghni Mkawi, Yas Al-Hadeethi, Bassim Arkook, Elena Bekyarova

**Affiliations:** 1Department of Physics, Faculty of Science, King Abdulaziz University, Jeddah 21589, Saudi Arabia; 2Center of Nanotechnology, King Abdulaziz University, Jeddah 42806, Saudi Arabia; 3Department of Chemistry, University of California at Riverside, Riverside, CA 92521, USA

**Keywords:** Cu_2_ZnSnS_4_ (CZTS) thin films, electrochemical deposition, 1,8-diiodooctane (DIO) additive, solar cell

## Abstract

Kesterite Cu_2_ZnSnS_4_ (CZTS) thin films using various 1,8-diiodooctane (DIO) polymer additive concentrations were fabricated by the electrochemical deposition method. The optical, electrical, morphological, and structural properties of the CZTS thin films synthesized using different concentrations of 5 mg/mL, 10 mg/mL, 15 mg/mL, and 20 mg/mL were investigated using different techniques. Cyclic voltammetry exhibited three cathodic peaks at −0.15 V, −0.54 V, and −0.73 V, corresponding to the reduction of Cu^2+^, Sn^2+^, Sn^2+^, and Zn^2+^ metal ions, respectively. The analysis of the X-ray diffraction (XRD) pattern indicated the formation of the pure kesterite crystal structure, and the Raman spectra showed pure CZTS with the A1 mode of vibration. Field emission scanning electron microscopy (FE-SEM) indicated that the films are well grown, with compact, crack-free, and uniform deposition and a grain size of approximately 4 µm. For sample DIO-20 mg/mL, the elemental composition of the CZTS thin film was modified to Cu:Zn:Sn: and S = 24.2:13.3:12.3:50.2, which indicates a zinc-rich and copper-poor composition. The X-ray photoelectron spectroscopy (XPS) results confirmed the existence of Cu, Sn, Zn, and S elements and revealed the element oxidation states. The electrochemical deposition synthesis increased the absorption of the CZTS film to more than 10^4^ cm^−1^ with a band gap between 1.62 eV and 1.51 eV. Finally, the photovoltaic properties of glass/CZTS/CdS/n-ZnO/aluminum-doped zinc oxide (AZO)/Ag solar cells were investigated. The best-performing photovoltaic device, with a DIO concentration of 20 mg/mL, had a short-circuit current density of 16.44 mA/cm^2^, an open-circuit voltage of 0.465 V, and a fill factor of 64.3%, providing a conversion efficiency of 4.82%.

## 1. Introduction

Photovoltaic cells based on kesterite Cu_2_ZnSnS_4_ (CZTS) are a promising technology owing to the characteristics of the material, which relate to desirable properties in thin-film photovoltaic technology [1,2,3]. The CZTS compound possesses characteristic optical properties that make it a promising photo-absorber material for solar cell application due to its excellent chemical and physical properties, high absorption region (α ≥ 104 cm^−1^) with a direct energy gap values between 1.0 eV and 1.6 eV in the visible light spectrum, p-type conductivity, and abundant and nontoxic components [4,5].

The efficiency of solar cells based on CZTS is still lower than that of copper indium gallium selenide thin-film photovoltaics, which can be explained by the lower phase purity of the formed thin film due to the formation of secondary phases [6]. Another substantial cause of reduced efficiency is disorder in the CZTS layer structure, where the Zn and Cu cation sites in the z = ¼, ¾ planes result in ZnCu and CuZn antisites [7]. This disorder in the CZTS layer causes optical bandgap variations, which cause drastic losses in the open-circuit voltage and, consequently, in the efficiency of the devices. It is therefore important to improve control over the growth parameters such as temperature, additives, complexing agents, and deposition methods.

CZTS thin films have been deposited with different methods, including co-sputtering [8], thermal evaporation [9], sol-gel [10], spin coating [11], pulsed laser [12], and electrochemical deposition [13].Among these methods, electrochemical deposition followed by annealing and sulfurization is the most promising fabrication method, considering the possibility for scale-up, ability to control the composition, and efficient use of raw material [14]. Two approaches are used in electrochemical CZTS thin film deposition: (i) electrodeposition in a single process, using either ionic liquids or aqueous electrolytes. In this instance, the main difficulty is in successfully leveling the reduction potentials of the predecessors, which have vastly differing standard potentials. Electrolyte composition is essential for achieving this objective since the correct precursor concentration and addition of appropriate complexing agents must be selected. In reality, the literature demonstrates that compositional heterogeneity and/or the existence of secondary phases hinder the production of a high-quality, one-step electrochemical CZTS thin film. To address this issue, (ii) two-step electrodeposition is used, in which the electrochemical deposition of CZTS thin film is followed by annealing at 500–600 °C and sulfurization with sulfur powder in the presence of nitrogen gas or H_2_S in the environment. This second procedure enhanced the crystallinity, surface morphology, light absorption, and electrical characteristics of CZTS thin films.

An efficiency of 7.99% has been recorded for the sulfurization of stacked electroplated CZTS thin film using Cu-poor and Zn-rich stoichiometry [15]. Despite the high control of the CZTS layer composition offered by the electrochemical deposition method, compositional deviations from stoichiometry due to secondary phase formation, element losses, and defects in the CZTS layers are still present.

The electrochemical deposition method is usually carried out with a chemical solution; this may lead to remaining unconsumed secondary phases due to the nonuniform mixing of cations. This problem can be resolved using an additive to improve the mixed cations and control the annealing and sulfurization more precisely. The most commonly used additive for organic photovoltaics (OPV) is 1,8-diiodooctane (DIO). DIO improves the phase separation morphology that results in power conversion efficiencies in bulk heterojunction (BHJ) OPVs [16].

In this work, we investigate metal (Cu, Zn, and Sn) thin film alloy fabricated by galvanostatic electrodeposition followed by annealing and sulfurization. We study the influence of DIO additive concentration on the compositional, structural, morphological, electrical, and optoelectronic properties of the CZTS active layer deposited on Mo substrates via different characterization techniques. This is the first time, to our knowledge, that a DIO additive has been used to improve CZTS thin film properties. The results indicate that the addition of DIO improves grain size uniformity and reduces defects, which directly affect the crystallization, optical, and morphological properties of CZTS thin film. We found that the addition of DIO can efficiently prevent carrier recombination on interfaces and enhance carrier collection due to the formation of large grain sizes. The use of DIO in CZTS thin-film fabrication will result in lower costs and higher power conversion efficiency for fabricated devices.

## 2. Experimental Section

In this method, metal layers of Cu–Zn–Sn (CZT) were deposited on soda–lime glass (SLG; 3.0 × 2.5 cm, 1 µm Mo/glass) via electrochemical deposition. The molybdenum-coated SLG (Mo-SLG) substrates were cleaned for 10 min by successive sonication in isopropyl alcohol, acetone, and, finally, deionized water DI for 10 min, respectively, to remove surface contamination. Aqueous solutions of oleylamin (total 40mL) were prepared that contained copper sulfate (CuSO_4_·5H_2_O—0.02 M), Zinc Sulphate 0.02 M (ZnSO_4_∙2H_2_O—0.02 M), tin sulfate (SnSO_4_—0.01 M). Sodium Thiosulfate Pentahydrate (Na_2_S_2_O_3_·5H2O—0.2 M) was used as complex agents and DIO additive concentrations of 5, 10, 15, and 20 mg/mL were used as the additive. To maintain the pH of the CZTS solution at approximately 5, 1 M tartaric acid (C_6_H_6_O_6_) was added to the electrolytic bath. Cyclic voltammetry (CV) was conducted at a scan rate of 10 mV/s using a saturated Ag/AgCl reference electrode, a Mo-SLG substrate as the working electrode, and a Pt mesh as the counter electrode. The distance between the counter and the working electrode was limited to 2 cm. The electrodeposition of CZTS films was executed using our innovation, the triangle wave pulse method; more details about this method can be found in our previous work [17].

After the deposition of these elements, the precursors were sulfurized by annealing with 200 mg sulfur powder at temperatures of 580 °C for 2 h in three zones of a quartz tube furnace. The samples were left to cool naturally. The samples were etched for 30 s in a 10% solution of potassium cyanide (KCN) and rinsed in distilled water. To complete the thin-film photovoltaic devices, an 80 nm CdS (n-type) buffer layer was deposited on the CZTS thin film (~1 µm thickness) using the chemical bath deposition method. The aqueous solution of deionized water containing 0.004 M cadmium sulfate (CdSO_4_), thiourea (0.03 M), and ammonia (20%) (1.5 M) heated up to 80 °C. After the CdS layer deposition, the CZTS/CdS films were washed with DI water and annealed at 180 °C for 15 min. Following that, an 80-nm-thick intrinsic zinc oxide (i-ZnO) as a window layer and aluminum-doped zinc oxide (AZO) thin films (~300 nm) as a transparent conducting oxide layer were successively deposited on the CZTS/CdS films using Rf magnetron sputtering of 100 W under a working pressure of ~10^6^ Pa with Ar as the inlet gas flow. Then, the samples were annealed at 350 °C for 20 min under N_2_ gas flow in a tube furnace. Finally, the SLG/Mo/CZTS/CdS/ZnO/AZO photovoltaic structure was completed by depositing 250-nm-thick Ag films by thermal evaporation.

Crystal structure properties of the obtained CZTS active layer were recorded by XRD using a Philips X-Ray Diffractometer (PW 3710) with an X-ray tube, CuKα radiation: λ = 1.5406 Å, 40 kV, and 30 mA. Raman scattering measurements of the CZTS thin film were carried out with an HR 800 micro-Raman system at 442 nm (2.8 eV) from a He–Cd laser. Morphologies and compositions of the CZTS thin films were examined by a field-emission scanning electron microscope (FE-SEM), Model JSM-6701F, UK, equipped with an EDS (EDS; model: EMAX 6853-H, Horiba, 20 kV, 11 μA). The chemical status and bonding were determined by XPS using aVG Multilab 2000, Thermo VG Scientific, UK. The optical properties of the CZTS films were studied using a UV–Vis spectrophotometer (Hitachi U-2900). The photovoltaic cells’ parameters were measured using a Keithley 2400 source meter under a simulated AM1.5G (100 mW/cm^2^).

## 3. Results and Discussion

Cyclic voltammeters (CVs) were obtained at room temperature in the voltage range between −1.0 and 1.0 V with a scan rate of 10 mV/s using a potentiostat with a three-electrode system to study Cu, Sn, and Zn reduction peaks as displayed in Figure 1. The first reduction peak was detected at −0.15 V vs. Ag/AgCl for the Cu reduction peak. The standard reduction potential for Cu ions was 0.16 V vs. Ag/AgCl in 1 M aqueous solution at 25 °C (Cu^2+^ + 2e^−^↔ Cu, *E^0^* = 0.15 V) [18]. At more negative potentials (>−0.50 V vs. Ag/AgCl), the current reached a maximum, which revealed the hydrogen evolution reaction overpotential. Similarly, anodic peaks were detected in the anodic scan due to the stripping of the metals. The Sn reduction peak occurred at −0.54 V [19]. The standard reduction potential of Sn ions was −0.52 V vs. Ag/AgCl in 1 M aqueous solution at 25 °C (Sn^2+^ + 2e^−^↔Sn, *E*^0^ = −0.54 V). The CV curve showed the Zn reduction peak at −0.73 V. The standard redaction potential of Zn was −0.76 V vs. Ag/AgCl in 1 m aqueous solution at 25 °C; Zn^2+^ + 2e^−^↔Zn, *E*^0^ = −0.76 V) [20].

XRD is a useful tool for analyzing the crystalline phase of the sample. Figure 2 displays the XRD patterns for the CZTS thin films fabricated using different DIO additive concentrations after annealing and sulfurization at 580 °C for 120 min. The samples show diffraction peaks at 2θ = 28.47°, 31.89°, 47.59°, and 56.24°, attributed to the (112), (200), (220) and (312) crystallographic planes of CZTS (JCPDS #26-0575), which verified the kesterite phase of the CZTS samples [21,22]. The diffraction peak at 2θ = 40.13° is related to the Mo back contact [23]. No other peaks were detected, which indicates that no impurities or secondary phases exist. The sharp and strong peak intensities were influenced by the DIO additive, which positively affects the crystal structure of the film, so that the increase in DIO concentration led to increased crystallinity of the film. The lattice parameters were calculated to be a = 0.5606 nm and c = 1.0204 nm, which agree well with values from our previous work (a = 0.5426 nm and c = 0.0847 nm) for CZTS material [24].

To investigate the influence of different additives on the crystal structure, Scherrer’s equation was used to calculate the crystallite size (D) of the CZTS films [25].
D = 0.9 λ βcos θ (1)
where β is the angular full width at half maximum (FWHM), θ is the diffraction angle, λ (1.5406 Å) is the X-ray wavelength, and K is a constant, usually set at K = 0.9. The average crystal size was found to be 29.34 nm for DIO-5 mg/mL, 34.76 nm for DIO-10 mg/mL, 37.38 nm for DIO-15 mg/mL, and 41.76 nm for DIO-20 mg/mL, respectively. These results indicate that the crystallite size increased with increasing DIO additive concentration, which led to increased sample crystallinity. Our experimental results (not included) showed that the increase in DIO additive concentration over 20 mg/mL affected solution viscosity and led to the formation of undissolved black agglomerates at the bottom of the beaker.

A better understanding of phase purity in the near-surface region was investigated by Raman spectroscopy, allowing the recognition of secondary phases that are common on the CZTS layer. Some secondary phases are comparable to CZTS phases, such as Cu_2_SnS_3_ or ZnS phases. The Raman spectroscopy was used to distinguish these phases. Raman spectra of CZTS films prepared using different DIO additive concentrations are shown in Figure 3. Raman spectra of CZTS samples display the kesterite structure, as confirmed by the two strong peaks located at 288 cm^−1^ and 338 cm^−1^ and the less intense peak at 306 cm^−1^ [26]. The strongest peak at 338 cm^−1^ indicates the A = A1 mode, which is linked to the vibrations of the sulfur atoms [27]. The peak at 285 cm^−1^ is assigned to the B transverse optical longitudinal optical (TO LO)/E (TO LO) mode [28]. The conformity of the Raman and the XRD peaks prove the absence of secondary phases, such as CTS, that have a peak at 314 cm^−1^, SnS with a peak at 150 cm^−1^, and CuS with a peak at 465 cm^−1^. The absence of the secondary phases may be because all secondary phases were consumed during thin-film growth with the assistance of the DIO additives. Additionally, the increase in additive concentrations increased the intensities of the Raman peaks of the CZTS thin film, thus significantly improving the crystal structure properties of the CZTS thin film.

Figure 4 displays the FE-SEM images of CZTS thin films prepared using different DIO additive concentrations and annealed and sulfurized at 580 °C for 2 h. Figure 4a shows the surface morphology of the DIO-5 mg/mL thin film, which has irregular grain-size distributions on a nonuniform surface with cracks and holes. Figure 4b shows that the DIO-10 mg/mL thin film had a tighter and nonhomogeneous structure, but the grain size was bigger than those in Figure 4a, which was approximately 2 µm. When the DIO additive concentration was 15 mg/mL, as shown in Figure 4c, the surface grain size increased most clearly, and compaction, uniformity, and smoothness improved considerably, suggesting that the additive had a significant influence on grain growth. The increase in grain size led to reduced grain boundaries (GBs),which improved the device’s performance. Figure 4d shows that the average grain size increased to exceed 4 µm with the DIO-20 mg/mL sample. The film was smooth, of uniform density, and free of cracks. The increase in GBs and the presence of defects would accumulate charge carriers, which would reduce the live time of carriers and increase the probability of charge recombination.

The chemical compositions of CZTS thin films prepared using different DIO additive concentrations were investigated using EDS analysis. Figure 5 shows the chemical compositions of CZTS thin films prepared using the DIO additive concentration of 20 mg/mL. The EDX spectra shown in Figure 5 confirm the presence of all predictable constitutive elements of CZTS. For the sample with 20 mg/mL of DIO, the atomic percent ratio of Cu: Zn:Sn:S was 24.2:13.3:12.3:50.2, which indicates that the material was copper-rich but close to the Cu:Zn:Sn:S = 2:1:1:4 ratio, confirming the formation of the stoichiometric Cu_2_ZnSnS_4_ composition.

Table 1 lists the elemental composition of the annealed CZTS films fabricated with different DIO additive concentrations. The compositions in Table 1 indicate that with an increase in additive concentrations, the composition became more stoichiometric. The compositions of the CZTS active layers in Table 1 also have a chemical composition Cu-poor and Zn-rich stoichiometry. The Cu/(Zn + Sn) ratio of the CZTS thin films decreased as the DIO additive concentration increased. The DIO-20 mg/mL sample showed that the Zn/Sn ratio was greater than 1, which is preferred to obtain the desired stoichiometric composition for high photoconversion in photovoltaic applications [29,30].

XPS is a surface analysis technique that was used to investigate the elements that exist on the surface of materials and the oxidation states of elements in CZTS samples. Figure 6 displays the XPS spectra of a CZTS active layer fabricated using a DIO additive concentration of 20 mg/mL. The Cu 2p core-level spectrum (Figure 6a) clearly displays the existence of Cu 2p 3/2 and Cu 2p 1/2 contributions concentrated at binding energy values of 934.24 eV and 953.97 eV, respectively, with a peak splitting of 19.73 eV, which indicates the existence of the Cu^+^ state [31]. In the XPS spectrum for Zn 2p (Figure 6b), the strong peaks were centered at binding energies of 1025.0 eV and 1042.29 eV, which were assigned to Zn 2p 1/2 and Zn 2p 3/2, respectively, with a peak separation of 17.17 eV, consistent with Zn^2+^ [32]. Figure 6c indicates that the binding energies for the Sn3d3/2 and Sn3d1/2 are placed at 486.52 eV and 495.02 eV, respectively, with a peak separation of 8.50 eV, confirming the Sn^4+^ oxidation state [33]. The S 2p core spectrum of “S” (Figure 6d) displays the peaks for 2p 3/2 and 2p 1/2 at 159.91 eV and 165.45 eV, respectively, with a peak splitting at 5.54 eV that confirms the S^2−^ state [34]. Overall, the core-level base elements of Cu, Zn, Sn and S are well proportionate to their locations for Cu^+^, Zn^2+^, Sn^4+^, and S^2−^ valence states, which confirms that the synthesized CZTS thin films are well matched to the stoichiometric formula of CZTS material.

To examine the compositional and structural uniformity, the composition distribution of the DIO-20 mg/mL sample cross-section was investigated using EDS composition mapping. Figure 7 shows the elemental mapping for S, Cu, Zn, and Sn of the CZTS thin film. All compositional elements were distributed uniformly and homogenously through the thin film.

To investigate the effect of DIO additive concentrations on the optical properties of the synthesized CZTS absorber layers, absorption (α) measurements were achieved for CZTS thin films in the 250–500 nm wavelength range, as shown in Figure 8. The additive concentration had a considerable effect on the absorption of the films. Figure 8 indicates that the absorption increased to more than 10^4^ cm^−1^ with the increase in additive concentration at visible wavelengths (400–800 nm). The energy bands of the fabricated CZTS films were determined by a plot of (αhν)^2^ versus photon energy (hν) of CZTS films, as displayed in Figure 8, which is based on the Tauc relation [35], Equation (2)
αhv = A(hv − Eg)^n^(2)
in which Eg is the energy gap, h is Planck’s constant, A is a parameter vector depending on the likelihood of transition, and n depends on the electronic transition, where n = 1/2 for direct allowed transitions and n = 2 for an indirect allowed transitions, for example. The energy gaps were calculated to be 1.62 eV, 1.61 eV, 1.55 eV, and 1.51 eV for samples DIO-5 mg/mL, DIO-10 mg/mL, DIO-15 mg/mL, and DIO-20 mg/mL, respectively. This reveals that the energy gap reduced as the DIO additive concentration increased. These energy gap values agree with those reported in previous studies [36,37]. The decrease in the energy gap value may be due to the amorphous nature of the material, with an increased disorder degree for thin films fabricated with lower additive concentrations. For the DIO-20 mg/mL sample, the energy gap was calculated to be 1.51 eV, which is considered the optimum value for an absorber layer that can be used to investigate a high-performance photovoltaic device. In addition, this optimum energy gap is achieved along with increased crystallinity and grain size, which increase the crystallite size and improve the light absorption ability in solar cells.

A FE-SEM cross-section photo of the device based on the sample DIO-20mg/mL (Figure 9) is taken to investigate the grain size and quality of the contacts at the interfaces. In the realization of the SLG/Mo/CIGS/CdS/i-ZnO/AZO/Ag structure, the image clearly shows that the CZTS thin film had an average size of 1 µm, few cracks and voids, and good adherence to the surface of the Mo substrate. It is difficult to detect the MoS_2_-resistant layer in the cross-section image, indicating that the additive and the method that were used prevented the formation of a MoS_2_ layer. Figure 9 shows the 80-nm-thick n-type CdS film coated via chemical bath deposition diffused into the 100 nm i-ZnO film that was deposited using Rf sputtering. The CdS buffer layer in the photovoltaic device inhibits shunt paths, reducing the number of interface defects that act as recombination centers. Finally, a 120-nm-thick metallic layer of aluminum-doped zinc oxide, ZnO: Al(AZO), which acts as a window layer, was deposited onto the CdS/n-ZnO layer using Rf sputtering.

To investigate the photovoltaic performance of the SLG/CZTS/CdS/n-ZnO/AZO/Ag structure, current density-voltage (J–V) characteristics were measured under standard AM 1.5 G illumination (100 mW/cm^2^), as shown in Figure 10. The solar-cell performance was specified by parameters such as the open-circuit voltage (V_oc_), short-circuit current density (J_sc_), fill factor (FF), and power conversion efficiency (η) which can be obtained from the curve shown in Figure 10. The shunt resistance (R_sh_) and series resistance (R_s_) performances of CZTS solar cells deduced from the J–V curves for samples prepared using different DIO additive concentrations are compared in Table 2. As illustrated in Figure 10, the short-circuit current density increased as the DIO additive concentration increased. The best-performing solar cell, prepared using a DIO concentration of 20 mg/mL, exhibited a J_sc_ of 16.44 mA/cm^2^, a V_oc_ of 0.465 V, and a FF of 64.3%, which resulted in an energy conversion efficiency of 4.82%. The increases in FF and overall cell efficiency may be due to the drop in the series resistance (R_s_) from 23.37 Ω cm^2^ to 14.27 Ω cm^2^, which can be explained by the improved photo-generated electron transport in the junction zone [38]. The short-circuit current density (J_sc_) rose from 11.74 mA/cm^2^ to 16.44 mA/cm^2^, which may be attributed to the CZTS thin films enhanced light absorption. The increase in V_oc_ from 432 mV to 456 mV may be attributable to the absence of defect states and the decrease in series resistance R_s_.

The improvement in cell efficiency obtained in this work (see Table 2) may be because of the improved quality of the CZTS active layer surface morphology (fewer cracks and longer carrier diffusion lengths), the grain size, and the crystallite, as well as the quality of the layered component interfaces, which led to the high photo-generated carrier transport capability of the layered components and optimum ratios of Cu/(Zn + Sn) and Zn/Sn.

## 4. Conclusions

Kesterite CZTS thin films have been successfully deposited on a Mo-SLG substrate by electrochemical deposition followed by annealing and sulfurization of the precursors at temperatures of 580 °C. The influences of different DIO additive concentrations (5 mg/mL, 10 mg/mL, 15 mg/mL, and 20 mg/mL) on the CZTS thin film properties were investigated. The CV results indicated that cathodic peak potentials for Cu^2+^, Sn^2+^, and Zn^2+^ ion reduction processes were observed at −0.15 V, −0.54 V, and −0.73 V, respectively. XRD studies showed that increasing additive concentration led to increased peak intensity, which indicated improvement in the crystallinity of the samples. The Raman spectra indicated the absence of secondary phases and confirmed kesterite formation and the single-phase structure of the samples. High absorption coefficients of more than 10^4^ cm^−1^ were achieved for the CZTS thin films. The optical direct energy gaps of the annealed CZTS thin films ranged from 1.62 eV for DIO-5 mg/mL to 1.51 eV for DIO-20 mg/mL, close to the optimum energy gap for a photovoltaic absorber layer. The EDS measurement confirmed that the formation of Cu-poor and Zn-rich CZTS samples becomes more stoichiometric. The best-performing SLG/CZTS/CdS/i-ZnO/AZO/Ag solar cell with DIO-20 mg/mL exhibited a short-circuit current density of 16.44 mA/cm^2^, an open-circuit voltage of 0.465 V, and a FF of 64.3%, providing an energy conversion efficiency of 4.82%.The improvement in energy conversion efficiency was observed as the additive concentration increased, which is attributed to the optimum optical energy gap, compositional ratio, smooth and compact surface morphology, and enhanced crystal quality. This work provided crucial information on the DIO additive and its role in making kesterite CZTS thin film a good absorber for solar cell applications.

## Figures and Tables

**Figure 1 materials-16-01659-f001:**
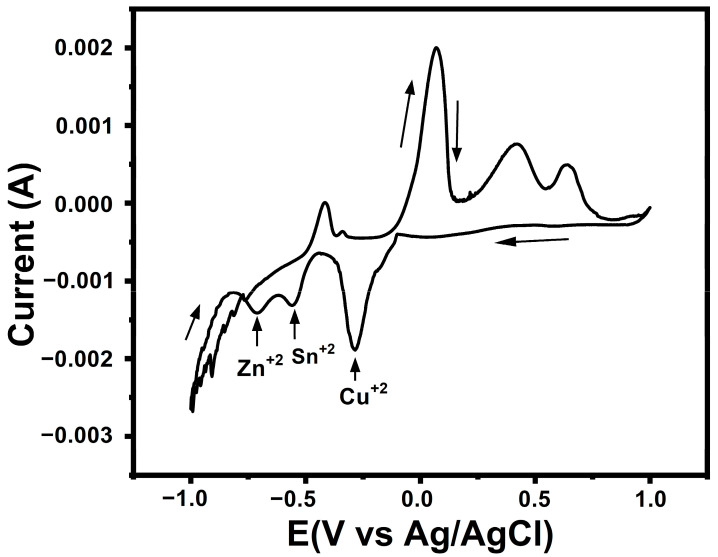
Cyclic voltammetry (CV) with scan speed10 mVs^−1^; with the calculated reduction potentials of Cu, Zn, and Sn.

**Figure 2 materials-16-01659-f002:**
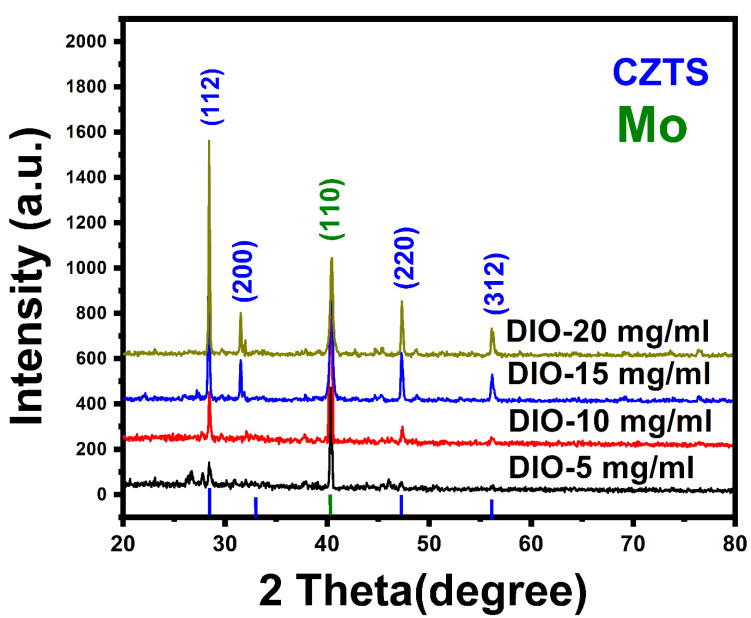
X-ray diffraction patterns of Cu_2_ZnSnS_4_ (CZTS) samples annealed at 580 °C for 120 min and sulfurized with 200 mg S prepared using different 1,8-diiodooctane (DIO) additive concentrations.

**Figure 3 materials-16-01659-f003:**
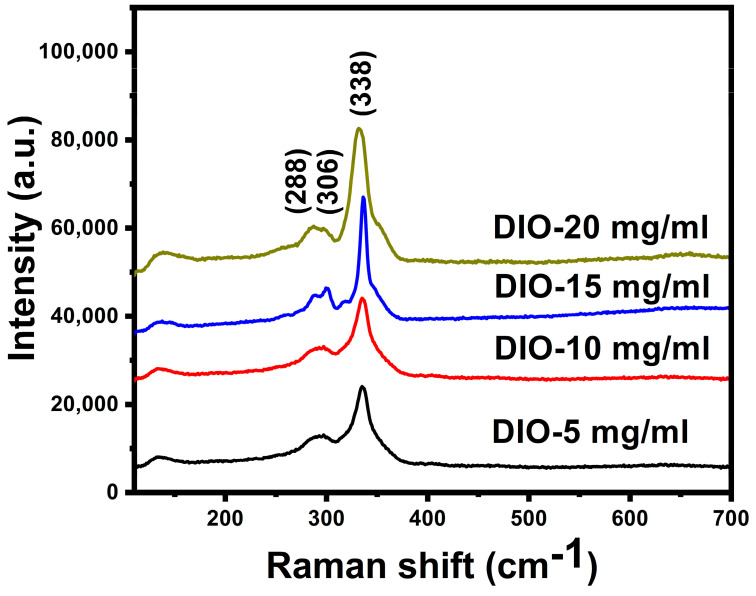
Raman spectra of Cu_2_ZnSnS_4_ (CZTS) absorber films in the wavenumber region of 100–700 cm^−1^, were prepared using prepared using different 1,8-diiodooctane (DIO) additive concentrations.

**Figure 4 materials-16-01659-f004:**
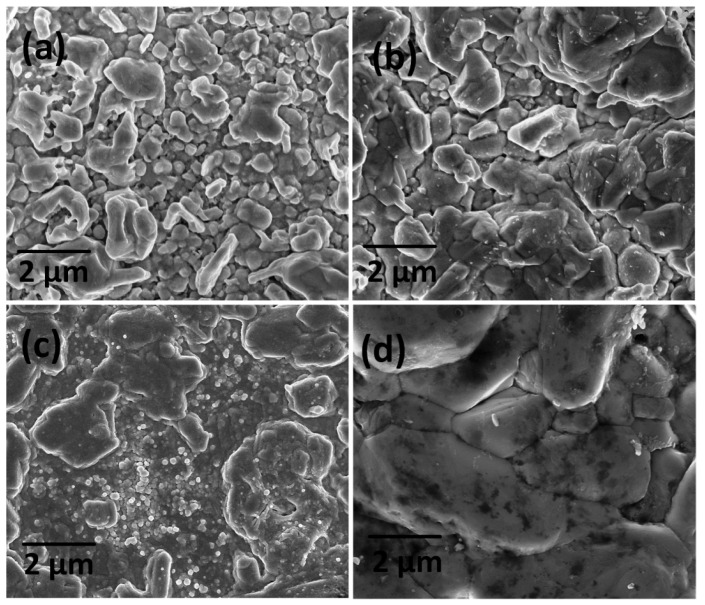
Field emission scanning electron micrographs of the Cu_2_ZnSnS_4_ (CZTS) layer surfaces prepared using 1,8-diiodooctane (DIO) additive concentrations of (**a**) 5, (**b**) 10, (**c**) 15, and (**d**) 20 mg/mL.

**Figure 5 materials-16-01659-f005:**
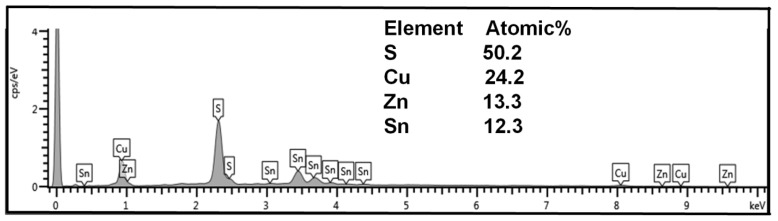
Energy-dispersive X-ray spectroscopy of the sample DIO-20 mg/mL, showing the elemental composition.

**Figure 6 materials-16-01659-f006:**
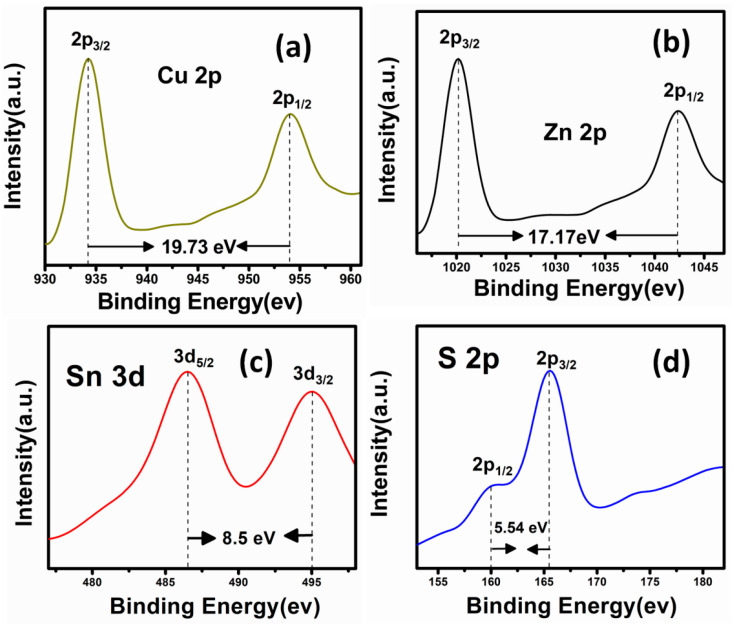
X-ray photoelectron spectroscopy (XPS) core level spectrum of (**a**) Cu 2p, (**b**) Zn 2p, (**c**) Sn 2p, and (**d**) S 2p peaks measured for an atypical CZTS thin film prepared using 20 mg/mL of 1,8-diiodooctane (DIO).

**Figure 7 materials-16-01659-f007:**
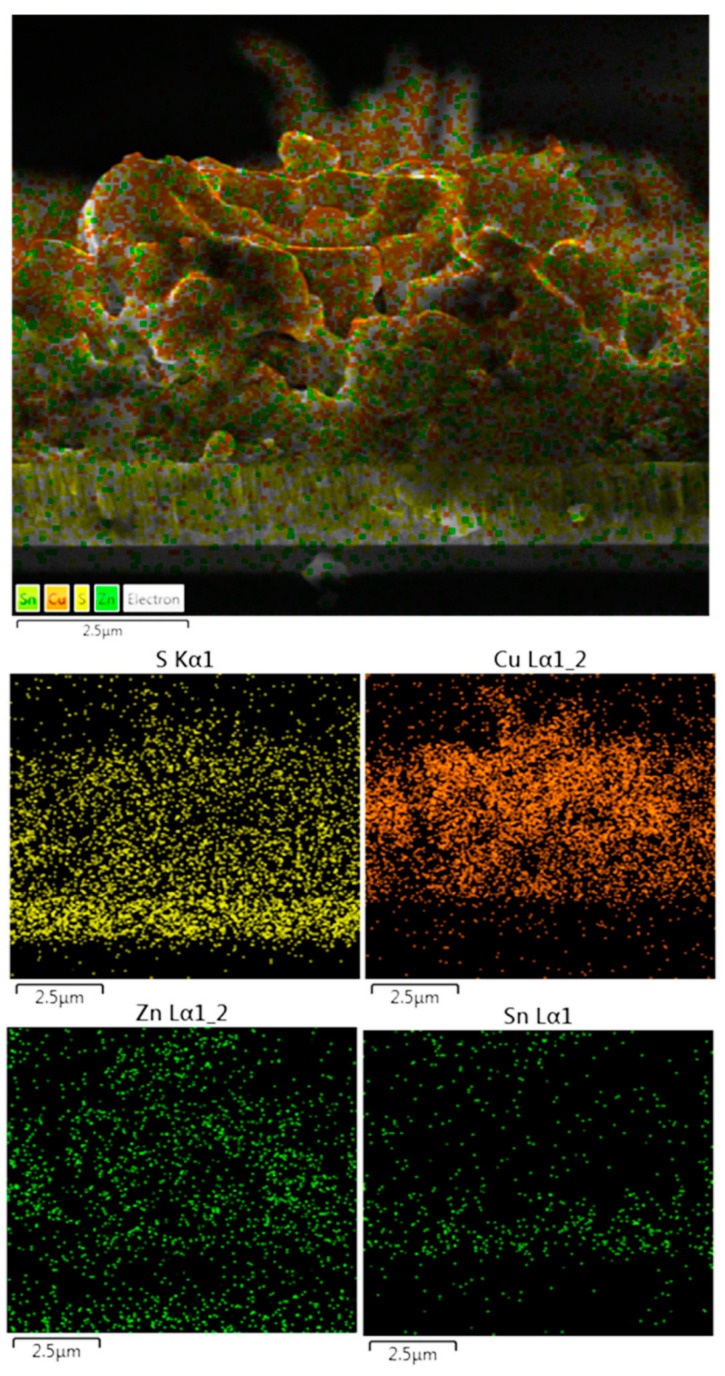
Microstructure evolution and energy dispersive spectroscopy (EDS) composition mapping of the DIO-20 mg/mL thin-film cross-section.

**Figure 8 materials-16-01659-f008:**
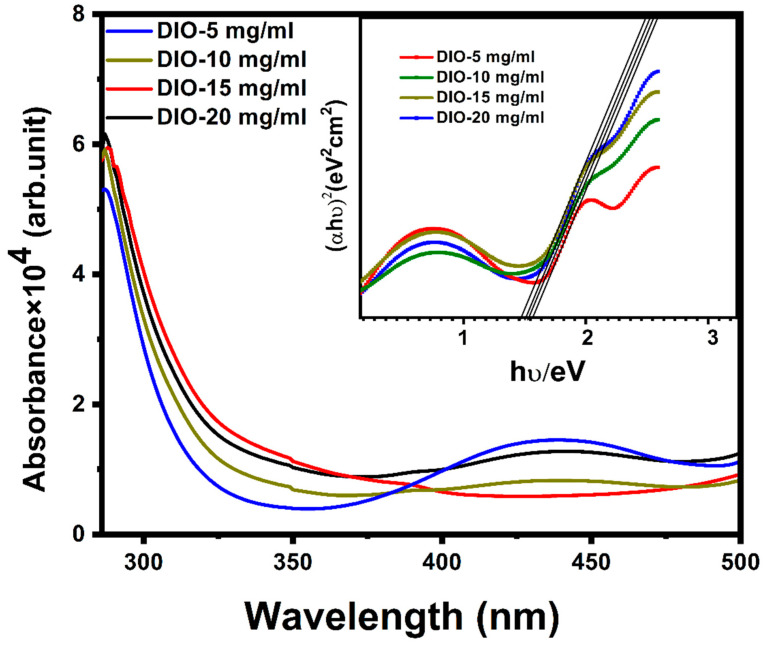
Absorption spectra of the typical Cu_2_ZnSnS_4_(CZTS) thin films and (inset) Tauc plots ((αhν)^2^ vs. hν)) for the determination of the energy gap.

**Figure 9 materials-16-01659-f009:**
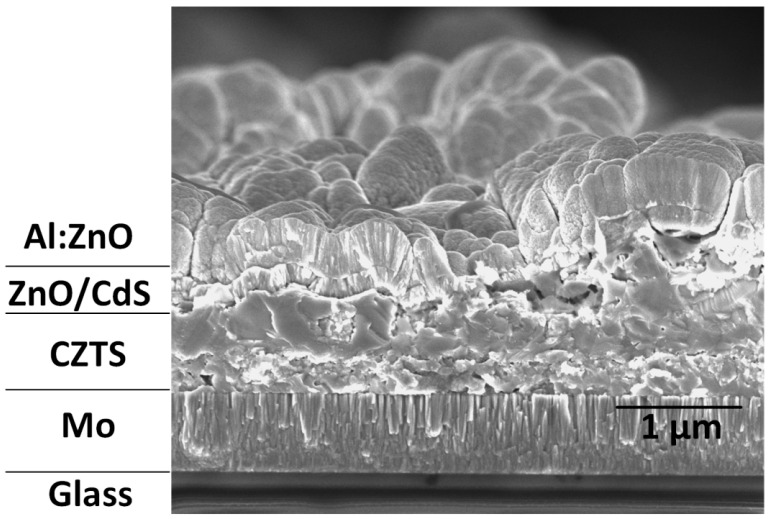
Field-emission scanning electron microscopy of the cross-section for the soda-lime glass (SLG)/Cu_2_ZnSnS_4_ (CZTS)/CdS/i-ZnO/aluminum-doped zinc oxide (AZO) photovoltaic device based on the sample DIO-20 mg/mL.

**Figure 10 materials-16-01659-f010:**
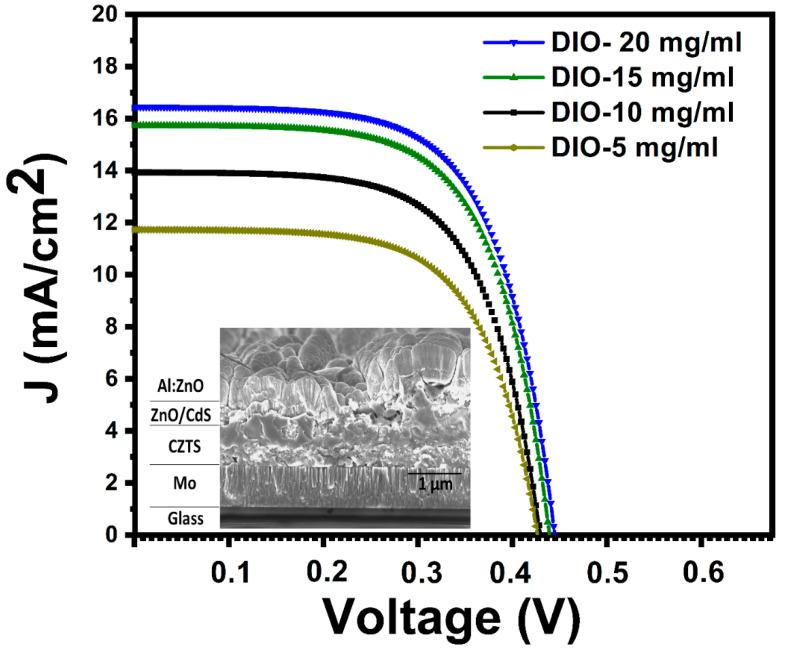
Current density-voltage (J–V) characteristic of a CZTS photovoltaic device, the inset shows a cross-sectional SEM micrograph of the photovoltaic.

**Table 1 materials-16-01659-t001:** Elemental composition of Cu_2_ZnSnS_4_ (CZTS) thin films fabricated using different 1,8-diiodooctane (DIO) additive concentrations.

Sample	Cu (at%)	Zn (at%)	Sn (at%)	S (at%)	Cu/(Zn + Sn)	Zn/Sn
DIO-5 mg/mL	26.5	12.8	13.2	48.3	1.03	0.97
DIO-10 mg/mL	25.6	12.9	12.9	48.5	0.99	1.00
DIO-15 mg/ml	24.7	13.1	12.5	49.6	0.96	1.04
DIO-20 mg/mL	24.2	13.3	12.3	50.2	0.94	1.08

**Table 2 materials-16-01659-t002:** Photovoltaic performance of the CZTS solar devices prepared using different 1,8-diiodooctane (DIO) additive concentrations.

Sample V_oc_ (mV)	J_sc_ (mA/cm^2^) FF (%)	η (%) R_s_ (Ω cm^2^)	R_sh_ (Ω cm^2^)
DIO-5 mg/mL 432	11.74 61.3	2.36 23.37	443.2
DIO-10 mg/mL 433	14.35 63.3	3.84 21.44	635.6
DIO-15 mg/mL 445	15.88 64.1	4.52 16.25	774.4
DIO-20 mg/mL 456	16.44 64.3	4.82 14.27	958.6

## Data Availability

The data presented in this study are available from the corresponding author, Elmoiz Mkawi, upon reasonable request.

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
