# Peer review of "Impact of 1,8-Diiodooctane (DIO) Additive on the Active Layer Properties of Cu2ZnSnS4 Kesterite Thin Films Prepared by Electrochemical Deposition for Photovoltaic Applications"

_materials, 2023, doi:10.3390/ma16041659_

Round 1

Reviewer 1 Report

1)      The authors are advised to use the accepted term “photovoltaic cell” instead of “Photovoltage cell”.

2)      Proof read of the manuscript is strongly required. There are many typo and grammatical errors. For example, DOI instead of DIO in the abstract; the use of “desirable properties” twice in the first sentence, the word high is given with capital H in the first paragraph; Oley amine and many more...  In many places there are unnecessary (extra) spaces between words (e.g. “..acetone, ___ and finally deionized..”), and in some cases there is no space between a word and comma (e.g. “copper sulfate (CuSO4.5H2O- 0.02 M ),Zinc Sulphate”). The use of subscript when presenting chemical formula was omitted. The authors must use a single font in the text.

3)      The authors should indicate the voltage range of cyclic voltammetry. What was the point in doing CV?

4)      The authors are required to present some details of the electrochemical deposition of CZTS.

5)      Where is the XRD peak claimed at 76.44 degree in the text?

6)      The authors should explain the improvement in Voc, Jsc with increased DIO concentration.

7)      The authors must use a proper, accepted referencing style. For example, [5] H. Yang, J. Zhang, M. Li, R. Hao, G. Song, (2018) is not acceptable.

Author Response

Response to Reviewers

Manuscript ID: materials-2188410
Title: Impact of 1, 8-diiodooctane (DIO) additive on the active layer properties of Cu2ZnSnS4 kesterite thin films prepared by electrochemical deposition for photovoltaic applications

Journal of Materials

First of all, in my capacity as the corresponding author, I would like to express my sincere gratitude and appreciation to the reviewer for his/her valuable comments, without which this manuscript would not have been valuable. I am really greatly indebted to the reviewer and to the editor.

Comments made by reviewer #1

1)      The authors are advised to use the accepted term “photovoltaic cell” instead of “Photovoltage cell”.

A1  #: The reviewer is correct; we corrected this information in the revised manuscript. Thanks for this observation.

2)      Proof read of the manuscript is strongly required. There are many typo and grammatical errors. For example, DOI instead of DIO in the abstract; the use of “desirable properties” twice in the first sentence, the word high is given with capital H in the first paragraph; Oley amine and many more...  In many places there are unnecessary (extra) spaces between words (e.g. “..acetone, ___ and finally deionized..”), and in some cases there is no space between a word and comma (e.g. “copper sulfate (CuSO4.5H2O- 0.02 M ),Zinc Sulphate”). The use of subscript when presenting chemical formula was omitted. The authors must use a single font in the text.

A2  #: The reviewer is correct; we corrected these typographical and grammatical errors in the revised manuscript. Thanks for this observation.

3)      The authors should indicate the voltage range of cyclic voltammetry. What was the point in doing CV?

A3  #:The reviewer is correct, we added this information in the revised manuscript, please see line 3 page 5. Thanks for this observation

- Cyclic voltammetry (CV) is a technique used to study reaction mechanisms that involve the transfer of electrons. We used CV to determine reduction voltages for the metals (Cu, Sn, and Zn) that were applied to deposit these metals during the electrochemical deposition process. In our work, CV indicated the Cu reduction peak at  −0.15 V, the Sn reduction peak at −0.54 V, and the Zn reduction peak at −0.73 V. This means that to deposit these metals, we should apply the voltage between -1.0 and 1.0 V vs. Ag/AgCl.

4)      The authors are required to present some details of the electrochemical deposition of CZTS.

A4  #:The reviewer is correct; we added this paragraph in the revised manuscript , please see line 19 page 2

“ Two approaches are used in electrochemical CZTS thin film deposition: (i) electrodeposition in a single process, using either ionic liquids or aqueous electrolytes. In this instance, the main difficulty is in successfully leveling the reduction potentials of the predecessors, which have vastly differing standard potentials. Electrolyte composition is essential for achieving this objective since the correct precursor concentration and addition of appropriate complexing agents must be selected. In reality, the literature demonstrates that compositional heterogeneity and/or the existence of secondary phases hinder the production of a high-quality, one-step electrochemical CZTS thin film. To address this issue, (ii) two-step electrodeposition is used, in which the electrochemical deposition of CZTS thin film is followed by annealing at 500–600 oC and sulfurization with sulfur powder in the presence of nitrogen gas or H2S in the environment. This second procedure enhanced the crystallinity, surface morphology, light absorption, and electrical characteristics of CZTS thin films’’.

5)      Where is the XRD peak claimed at 76.44 degree in the text?

A5  #: we agree with the reviewer, we deleted this degree in the revised manuscript. it is an editorial mistake . Thanks for this observation

6)      The authors should explain the improvement in Voc, Jsc with increased DIO concentration.

A6  #:The reviewer is correct; we added this paragraph in revised manuscript , please see line 14 page 16

“The short-circuit current density (JSC) has risen from 11.74 mA/cm2 to 16.44 mA/cm2, which may be attributed to the CZTS thin films' enhanced light absorption. The increase in Voc from 432 mV to 456 mV may be attributable to the absence of defect states and the decrease in series resistance Rs’’.

7)      The authors must use a proper, accepted referencing style. For example, [5] H. Yang, J. Zhang, M. Li, R. Hao, G. Song, (2018) is not acceptable.

A7  #: The reviewer is correct; we revised all references in the revised manuscript. Thanks for this observation.

Reviewer 2 Report

The authors present the manuscript entitled "Impact of 1, 8-diiodooctane (DIO) additive on the active layer properties of Cu2ZnSnS4 kesterite thin films prepared by electrochemical deposition for photovoltaic applications". The paper itself is well written and in adequate order. However, many minor mistakes concerning the language and also order of the characters or letters where found. Therefore, a further revision of the manuscript is needed. A document including some errors can be found attached to the present revision.

In general the use of DIO to enhance the PV charter of the CZTS films is demonstrated and it can be followed correctly. However, some questions arise after te following comments:

From Figure 9 we can see the cross section of the deposited films.

The authors mention that a CZTS film of 2 microns can be seen. However, according to the micrograph and size bar, it seems to be less then a micron. 

Can you explain?

Also, it says "without any cracks or voids and with good adherence to the Mo substrate surface". From my point of view the film present even 2 zones and several voids. Can the authors comment about it?

Moreover, the ZnO/CdS film is very difficult to distinguish. Have the authors a better image of the zone?

All the other topics are quite clear and in general is a good work.

Author Response

Response to Reviewers

Manuscript ID: materials-2188410
Title: Impact of 1, 8-diiodooctane (DIO) additive on the active layer properties of Cu2ZnSnS4 kesterite thin films prepared by electrochemical deposition for photovoltaic applications

Journal of Materials

First of all, in my capacity as the corresponding author, I would like to express my sincere gratitude and appreciation to the reviewer for his/her valuable comments, without which this manuscript would not have been valuable. I am really greatly indebted to the reviewer and to the editor.

Comments made by reviewer #2

1-    From Figure 9 we can see the cross section of the deposited films.

The authors mention that a CZTS film of 2 microns can be seen. However, according to the micrograph and size bar, it seems to be less then a micron.

Can you explain?

A1  #: We agree with the reviewer; we corrected the CZTS film thickness to be 1µm in the revised manuscript. It is an editorial error , please see line 14 page 14, thanks for this observation.

Also, it says "without any cracks or voids and with good adherence to the Mo substrate surface". From my point of view the film present even 2 zones and several voids. Can the authors comment about it?

A2  #: The reviewer is correct; we corrected this information and added this paragraph to the revised manuscript. please see line 14 page 14

‘’ the image clearly shows that the CZTS thin film has an average size of 1µm, few cracks and voids, and good adherence to the surface of the Mo substrate’’.

Moreover, the ZnO/CdS film is very difficult to distinguish. Have the authors a better image of the zone?

A3  #: The reviewer is correct; the CdS layer was deposited using chemical bath deposition. The coherence of the CdS layer was low due to the method used to deposit this layer. The ZnO particles insert deeply into the CdS layer when the ZnO layer is deposited by RF spurting. For that reason, it is very difficult to distinguish the interface between the CdS and ZnO layers.

We appreciate all your efforts, your questions, feedback and suggestions that were extremely helpful to us.